# Gut Microbiota and Metabolome Changes in Three Pulmonary Hypertension Rat Models

**DOI:** 10.3390/microorganisms11020472

**Published:** 2023-02-13

**Authors:** Lingjie Luo, Haoyang Yin, Deming Gou

**Affiliations:** 1Shenzhen Key Laboratory of Microbial Genetic Engineering, Vascular Disease Research Center, College of Life Sciences and Oceanography, Shenzhen University, Shenzhen 518060, China; 2School of Pharmaceutical Sciences, Shenzhen University Health Science Center, Shenzhen University, Shenzhen 518060, China

**Keywords:** pulmonary hypertension (PH), gut microbiota, biomarker, metabolites

## Abstract

Dysbiosis of the gut microbiota and metabolites is found in both pulmonary hypertension patients and pulmonary hypertension rodent models. However, the exact changes in gut microbiota during the development of pulmonary hypertension is unclear. The function of the gut microbiota is also ambiguous. Here, this study showed that the gut microbiota was disrupted in rats with hypoxia (Hyp)-, hypoxia/Sugen5416 (HySu)-, and monocrotaline (MCT)-induced pulmonary hypertension. The gut microbiota is dynamically changed during the development of Hyp-, HySu-, and MCT-induced rat pulmonary hypertension. The variation in the α diversity of the gut microbiota in Hyp-induced pulmonary hypertension rats was similar to that in rats with MCT-induced pulmonary hypertension and different from that in rats with HySu-induced pulmonary hypertension. In addition, six plasma biomarkers, His, Ala, Ser, ADMA, 2-hydroxybutyric acid, and cystathionine, were identified in Hyp-induced pulmonary hypertension rats. Furthermore, a disease-associated network connecting *Streptococcus* with Hyp-induced pulmonary hypertension-associated metabolites was described here, including trimethylamine *N*-oxide, Asp, Asn, Lys, His, Ser, Pro, and Ile.

## 1. Introduction

Pulmonary hypertension (PH) is an incurable cardiovascular disease resulting from a progressive increase in pulmonary vascular resistance and elevated pulmonary arterial pressure. The characteristics of PH are vasoconstriction, vascular remodeling, inflammation, and thrombosis in situ [1]. Several vascular constriction drugs have been used to treat pulmonary hypertension by targeting the endothelin pathway, prostacyclin pathway, and nitric oxide (NO) signaling pathway [2,3,4,5]. However, these drugs cannot cure PH due to the other pathophysiologies of PH. Thus, it is important to explore new targets associated with the pathophysiology of PH.

Gastrointestinal tract tissues harbor hundreds of species of microbiota that intimately interact with hosts and provide them with genetic, metabolic, and immunological benefits [6]. The gut microbial community is changed and falls out of balance in multiple diseases such as cardiovascular disease [7], colorectal cancer [8], inflammatory bowel disease (IBD) [9], irritable bowel syndrome (IBS) [10], type I diabetes and type 2 diabetes [11], obesity, rheumatoid arthritis [12], and PH [13]. Alterations in the gut microbial community were observed in IPAH patients and chronic obstructive pulmonary disease (COPD) patients [13,14]. Alternative gut microbes were also observed in hypoxia/Sugen 5416-induced and monocrotaline (MCT)-induced PH rats [15,16]. However, these results only represent the change in gut microbiota in a given time period, and the exact changes in gut microbiota over the development of PH is unclear. It is also unclear which type of PH in humans is most similar to changes in the gut microbiota in the rat model.

Gut microbiota dysbiosis is involved in the development of diseases [17]. IBD patients have impaired lung function, whereas patients with COPD have increased intestinal permeability [17,18]. The gut microbiota plays a crucial role in influencing the development of host immunity. The pattern recognition receptors (PRRs) of immune cells in the gastrointestinal tract can recognize pathogen-associated molecular patterns (PAMPs) originating from the gut microbiota, including LPS (lipopolysaccharide) [19], exopolysaccharide [20], peptidoglycan [21], and RNA [22]. These immune responses maintain the homeostasis between the gastrointestinal tract and gut microbiota by activating immune cells to produce large proinflammatory cytokines such as interleukin 6 (IL-6) and interleukin-1β (IL-1β) [23]. However, the metabolites produced by gut microbiota are also involved in the pathophysiology of disease. Kaye’s group found that acetate supplementation significantly reduced systolic and diastolic blood pressures, cardiac fibrosis, and left ventricular hypertrophy [24]. The tryptophan metabolite indole-3-acetate is produced by the lung microbiota and suppresses macrophage inflammatory responses [25]. Vizcaino’s group indicated that both acetate in plasma and the acetate-producing bacteria *Parabacteroides* were reduced in hypoxia/Sugen 5416-induced PH rats [16]. Hansbro’s group indicated that *Streptococcus parasanguinis_B* is associated with the COPD-associated metabolite N-acetylglutamate and its analog N-carbamoylglutamate [14]. Koichiro Tatsumi’s group indicated that the development of PH was suppressed by antibiotic-induced modification of the gut microbiota [26]. This evidence indicates that an imbalance in the gut microbiota may play an important role in the pathogenesis of PH.

To determine the exact changes of gut microbiota over the development of PH, the intestinal microbial community in hypoxia-, hypoxia/Sugen5416-, and MCT-induced PH rats were analyzed using 16S rRNA sequencing. The results indicated that the gut microbiota was dynamically changed in the three PH rodent animal models. To find out the connection between gut microbiota and metabolites, the correlation analysis between the gut microbiota and metabolites were performed and an association within the gut–lung axis was found. The analysis of gut microbiota and metabolites not only sheds light on the active role of gut microbiota in the development of PH via the gut–lung axis, but also provides new targets for PH therapy.

## 2. Materials and Methods

### 2.1. Animals and Reagents

Eight-week-old male wild-type rats (weight, 200–220 g) (Guangdong Medical Laboratory Animal Center, Guangzhou, China) were used to establish models of hypoxia-, hypoxia/Sugen 5416-, and MCT-induced PH. Wild-type rats were housed in a barrier environment at 22 °C. All animals received humane care, and all procedures were approved by the Animal Care and Use Committee of Shenzhen University.

### 2.2. Establishment of Animal Models

Twenty rats were divided into four groups: the control group received PBS treatment (normoxia group, Nor, *n* = 5); the hypoxic group was exposed to hypoxia (10% O_2_) for four weeks (hypoxia group, Hyp, *n* = 5), and the hypoxia/Sugen 5416-induced group was subcutaneously injected with Sugen 5416 (10 mg/kg), exposed to hypoxia (10% O_2_) for three weeks, and then exposed to normoxia (20% O_2_) for two weeks (hypoxia/Sugen 5416 group, HySu, *n* = 5). The MCT group rats were subcutaneously injected with MCT (50 mg/kg) (MCT group, MCT, *n* = 5).

### 2.3. 16S rRNA Gene Amplification and Multiparallel Sequencing

Fecal samples originating from Nor, Hyp, HySu, and MCT rats were immediately frozen at −80 °C. Total genomic DNA from the samples was extracted. The 16S rRNA genes of the 16S V3–V4 regions were amplified using specific primers with barcodes. PCR was performed with Phusion High-Fidelity Taq Enzyme (NEB) following the manufacturer’s recommendations. The 16S V3–V4 regions were then purified with a DNA Gel Extraction Kit (Promega, Madison, WI, USA) following the manufacturer’s recommendations. An Ion Plus Fragment Library Kit (48 rxns; Thermo Scientific, Waltham, MA, USA) was used to generate sequencing libraries following the manufacturer’s recommendations. Library quality was assessed on a Qubit@ 2.0 Fluorometer (Thermo Scientific). Finally, the libraries were sequenced on the Ion S5 XL platform, and 400 bp/600 bp single-end reads were generated.

### 2.4. Metabolite Extraction

The plasma of Nor and Hyp rats was collected at sacrifice. Fifty microliters of plasma was added to 200 µL of methanol at 0 °C to suppress enzymatic activity. The extract solution was mixed with 150 µL Milli-Q water, and then 300 µL of the mixture was centrifugally filtered through a Millipore 5-kDa cutoff filter (ULTRAFREE MC PLHCC, HMT) for 2 h at 10,000× *g* and 4 °C. The filtrate was then evaporated to dryness under vacuum and reconstituted in 50 µL Milli-Q water for metabolome analysis at Human Metabolome Technologies, Inc. (HMT, Tsuruoka, Japan).

### 2.5. Metabolome Analysis

Metabolome analysis was conducted according to HMT’s Basic Scan package using capillary electrophoresis time-of-flight mass spectrometry (CE-TOFMS) based on previously described methods [27]. CE-TOFMS is a powerful method for profiling charged metabolites, because it provides high mass accuracy and an excellent resolution for the simultaneous measurement of metabolites in a wide continuous manner (50–1000 *m/z*) [28]. TOFMS is insufficient for metabolite identification, although it is good for mass accuracy. Briefly, CE-TOFMS analysis was carried out using an Agilent CE capillary electrophoresis system equipped with an Agilent 6210 time-of-flight mass spectrometer (Agilent Technologies, Inc., Santa Clara, CA, USA). The systems were controlled by Agilent G2201AA ChemStation software version B.03.01 (Agilent Technologies) and connected by a fused silica capillary (50 μm i.d. × 80cm total length) with commercial electrophoresis buffer (H3301-1001 and I3302-1023 for cation and anion analyses, respectively, HMT) as the electrolyte. The spectrometer was scanned from *m/z* 50 to 1000, and peaks were extracted using MasterHands automatic integration software (Keio University, Tsuruoka, Japan) to obtain peak information, including *m/z*, peak area, and migration time (MT). Signal peaks corresponding to isotopomers, adduct ions, and other product ions of known metabolites were excluded, and the remaining peaks were annotated according to HMT’s metabolite database based on their *m*/*z* values and MTs. Areas of the annotated peaks were then normalized to internal standards and sample amount to obtain the relative levels of each metabolite. Primary metabolites were absolutely quantified based on one-point calibrations using their respective standard compounds. Principal component analysis (PCA) was performed using R programs. The detected metabolites were plotted on metabolic pathway maps using VANTED software.

### 2.6. Hematoxylin and Eosin Staining

Lung tissues were fixed in formalin and embedded in paraffin (Leica, Weztlar, Germany), and 6 nm sections of lung tissues were deparaffinized and stained with hematoxylin and eosin as previously described [29].

### 2.7. Data Analysis

Uparse software (Uparse v7.0.1001, http://www.drive5.com/uparse/; accessed on 5 March 2021) was used to perform sequences analysis and cluster sequences with ≥97 % similarity of the samples to allow a selection of representative sequences in operational taxonomic units (OTUs). The OTU sequences were annotated based on Silva database (https://www.arb-silva.de/; accessed on 8 March 2021). To obtain the phylogenetic relationship of all OTU representative sequences, MUSCLE software (Version 3.8.31, http://www.drive5.com/muscle/; accessed on 20 March 2021) was used to perform rapid multiple sequence alignment. Finally, OTUs’ abundance information were normalized. Alpha and Beta diversity analysis were based on the normalized data.

Beta diversity analysis was used to evaluate differences of samples in species complexity; QIIME software (Version 1.7.0) was used to calculate weighted and unweighted unifrac of beta diversity. Cluster analysis was preceded by principal component analysis (PCA), which was applied to reduce the dimension of the original variables using the FactoMineR package and ggplot2 package in R software (Version 2.15.3). Principal Coordinate Analysis (PCoA) was performed to get principal coordinates and visualize from complex, multidimensional data. A distance matrix of weighted or unweighted unifrac among samples obtained before was transformed to a new set of orthogonal axes, by which the maximum variation factor is demonstrated by first principal coordinate, and the second maximum one by the second principal coordinate, and so on. PCoA analysis was displayed by WGCNA package, stat packages, and ggplot2 package in R software (Version 2.15.3).

### 2.8. Statistical Analysis

The results are expressed as the mean ± SEM from at least five rats and *p* < 0.05 was considered significant. Statistical significance was evaluated following normality testing with Shapiro–Wilk methods. When only two groups were compared, statistical differences were assessed with unpaired two-tailed Student’s *t* test if normally distributed. Statistical significance among three groups or over was determined using one-way analysis of variance (ANOVA) followed by Student–Newman–Keuls multiple comparison test. For non-parametric distributions, the Kruskal–Wallis test followed by the Dunnett post hoc test were performed. GraphPad Prism 9.0 (La Jolla, CA, USA) software was used to analyze the data and for graph generation.

## 3. Results

### 3.1. Animal Profiles

To explore the links between the gut microbiota and PH, we established Hyp-, HySu-, and MCT-induced PH rat models. Compared with the Nor group, the right ventricular systolic pressure (RVSP), right ventricular hypertrophy index (RV/[LV+S]), and vascular remodeling were increased in the three models (Figure 1A–D). The gut microbiome in Nor (*n* = 5), Hyp (*n* = 5), HySu (*n* = 5), and MCT (*n* = 5) rats were separately characterized by analyzing stool samples.

### 3.2. Overall Microbial Content of Nor, Hyp, HySu, and MCT Rats

To compare the gut microbiota of rats in the three PH models, prokaryotic 16S ribosomal RNA (rRNA) in the variable V3–V4 region of fecal samples from the Nor, Hyp, HySu, and MCT groups at weeks 0, 2, and 4 was sequenced. Based on the unique sequences of each sample and the specific filtering conditions, a total of 4,771,328 (1,201,820 for the Nor group, 1,201,873 for the Hyp-induced group, 1,173,714 for the HySu-induced group, and 1,193,921 for the MCT-induced group) high-quality clean reads were acquired from all the samples, with an average of 79,522 (range: 64,711–80,324) reads per sample used for downstream statistical analysis. All the sequences were clustered into 1063 OTUs. More than 99.5% of samples had good coverage, indicating sufficient community coverage. The details are shown in Appendix A.

Firstly, the alpha diversity (α-diversity) and beta diversity (β-diversity) within groups at Day 0 (week 0), Day 14 (week 2), and Day 26 (week 4) were compared. The Hyp group had decreased α-diversity (as measured by the Shannon and Simpson indices) at week 4 and increased richness indices (as measured by Chao1, observed species, and ACE) at week 2 and week 4 (Figure 2A). The HySu group had decreased α-diversity at week 2 and recovered to normal at week 4, and the richness indices were increased at week 4 (Figure 2B). The MCT group showed increased α-diversity at week 2, and the richness indices were increased at week 2 and week 4 (Figure 2C). These results suggested that the three PH model rats showed dynamic α-diversity values and richness indices at week 2 and week 4. The Nor group rats had decreased α-diversity (as measured by the Shannon and Simpson indices) at week 4, whereas the richness indices showed no change (Appendix A).

Then, the α-diversity and β-diversity among groups at Day 14 (week 2) and Day 26 (week 4) were compared. Compared with the Nor group, the HySu group had decreased α-diversity and richness indices at week 2, whereas the Hyp group had increased α-diversity (Figure 2D). At week 4, the α-diversity and richness indices were increased in the Hyp and MCT groups (Figure 2E).

The rarefaction curves of the observed species approached a plateau, indicating that the sequencing was sufficient in all samples and covered all the OTUs (Appendix A). The rank abundance curves fell slowly, indicating that the samples were not dominated by a few OTUs but mostly by low-abundance OTUs (Appendix A). Alterations in the microbiota composition of all the groups and samples were noted based on a PcoA (Appendix A). Additionally, principal component analysis (PCA) showed that the samples from the Nor-, Hyp-induced, HySu-induced, and MCT-induced groups were separated (Appendix A).

### 3.3. Alterations in the Gut Microbiota in Response to Hypoxia, Hypoxia/Sugen 5416, and MCT

To determine the changes in the gut microbiota that are associated with Hyp-induced, hypoxia/Sugen-induced, and MCT-induced PH, the differences of the gut microbiota were analyzed among the four groups. At the phylum level, compared with week 0, the abundance of *Firmicutes* was significantly increased, the abundances of *Actinobacteria*, *unidentified bacteria*, and *Verrucomicrobia* were significantly decreased, and the ratio of *Firmicutes/Bacteroidetes* was increased within the Hyp-induced group (Figure 3A,B). Compared with week 0, the abundance of *Firmicutes* was significantly increased, the abundances of *Bacteroidetes* and *unidentified bacteria* were significantly decreased, and the ratio of *Firmicutes/Bacteroidetes* was increased within the HySu-induced group at week 4 (Figure 3C). Within the MCT group, the abundances of *Firmicutes* and *Actinobacteria* were increased at week 4, and the abundances of *Bacteroidetes*, *unidentified bacteria*, and *Verrucomicrobia* were decreased at week 4 (Figure 3E,F). The ratio of *Firmicutes/Bacteroidetes* was increased within the MCT-induced group at week 4 (Figure 3E).

At week 2, compared with those in the Nor group, the abundances of gut microbiota have not significantly changed (Appendix A). At week 4, compared with those in the Nor group, the abundance of *Firmicutes* was significantly decreased, the abundances of *Bacteroidetes* and *Tenericutes* were significantly increased, and the ratio of *Firmicutes/Bacteroidetes* was decreased in the Hyp-induced group. The abundance of *Actinobacteria* was significantly increased in the MCT-induced group (Appendix A).

At the family level, *Lactobacillaceae*, *Bacteroidaceae*, *Prevotellaceae*, *Muribaculaceae*, and *Rikenellaceae* were the most abundant representatives of the phylum *Bacteroidetes*. *Lachnospiraceae*, *Peptostreptococcacea*, *Erysipelotrichaceae*, and *Ruminococcaceae* were the most abundant representatives of the phylum *Firmicutes*. The family of *unidentified Clostridiales* was the most abundant representative of the phylum *Proteobacteria*. Overall, compared with week 0, the abundances of *Peptostreptococcaceae* and *unidentified Clostridiales* were increased, and the abundances of *Bacteroidaceae* and *Prevotellaceae* were decreased in the Hyp-induced group at week 4 (Figure 4A,B). In the HySu-induced group, compared with week 0, the abundances of *Ruminococcaceae* and *Peptostreptococcaceae* were increased, and the abundances of *Lactobacillaceae* were decreased at week 4. In particular, the abundance of *Rikenellaceae* was decreased at week 2 (Figure 4C,D). In the MCT-stimulated group, the abundances of *Lactobacillaceae* and *Ruminococcaceae* changed at week 2, the abundances of *Peptostreptococcaceae* changed at week 4, and the abundance of *Erysipelotrichaceae* and *Prevotellaceae* changed at both week 2 and week 4 (Figure 4E,F). All of the microbiota showed dynamic changes among the groups (Appendix A–D).

To identify biomarkers for the Hyp-induced, hypoxia/Sugen-induced, and MCT-induced groups, the differences in microbial components among all the groups as well as those between selected groups were compared by linear discriminant analysis and linear discriminant effect size (LEfSe). When the gut microbiota of the Hyp group at weeks 0, 2, and 4 were analyzed together, 6 total discriminative features were identified. *Peptostreptococcaceae* of the phylum *Firmicutes*, *unidentified Clostridiales* of the phylum *Proteobacteria*, and *Romboutsia* were discriminative at week 4 in the Hyp-induced group (Figure 5A). When the gut microbiota of the HySu group at weeks 0, 2, and 4 were analyzed together, *Peptostreptococcaceae*, *Romboutsia*, *Lactobacillus*, *Lactobacillaceae*, *Lactobacillales*, and *bacilli* were discriminative at week 2. At week 4, *Lachnospiraceae*, *Ruminococcaceae*, and *Clostridiales* were discriminative (Figure 5B). When the gut microbiota of the MCT group at weeks 0, 2, and 4 were analyzed together, *Clostridiales*, *Bacteroidetes*, *Ruminococcaceae*, *Rikenellaceae*, *Muribaculaceae*, and *Alistipes* were discriminative at week 2. At week 4, *Peptostreptococcaceae*, *Romboutsia, Erysipelotrichaceae*, *Clostridiales*, and *Bacterides plebeius* were discriminative (Figure 5C).

When the Nor, Hyp, HySu, and MCT groups were analyzed together at week 2, unidentified *Lachnospiraceae* and *Candidatus Arthromitus* were identified in normoxic rats, and bacilli, *Lactobacillales*, *Lactobacillaceae*, *Fusicatenibacter*, and *Staphylococcus sciuri* were predominant in the HySu-induced group. *Clostridiales*, *Lachnospiraceae*, *Acetatifactor*, *Rikenellaceae*, *Alistipes*, and *Adiercreutzia* were predominant in the MCT-induced group (Figure 5D). At week 4, *Turicibacter*, *Erysipelotrichaceae*, *Erysipelotrichia*, and *Erysipelotrichales* were predominant in the MCT-induced group. Unidentified *Lachnospiraceae* were predominant in the HySu-induced group. *Coryhebacterum urealyticum*, *Bacteroidetes*, *Bacteroidles*, *Bacteroidia*, and *Muribaculaceae* were predominant in the Hyp-induced group (Figure 5E,F).

### 3.4. Potential Functions of the Gut Microbiota in Hypoxia-Induced, Hypoxia/Sugen 5416-Induced, and MCT-Induced Rats

To reveal the potential function of the gut microbiota in Hyp-induced, HySu-induced, and MCT-induced PH rats, a Kyoto Encyclopedia of Genes and Genomes (KEGG) analysis was performed. Appendix A shows the predicted pathways at level 2 (B) in the Hyp-induced, HySu-induced, and MCT-induced groups at weeks 0, 2, and 4. At level 2, the gut microbiota of the Hyp-induced group was mainly involved in transcription, cellular processes and signaling, xenobiotics biodegradation and metabolism, carbohydrate metabolism, metabolism of other amino acids, lipid metabolism, and glycan biosynthesis and metabolism. The gut microbiota of the HySu-induced group was mainly involved in nucleotide metabolism, replication and repair, immune system, nervous system, amino acid metabolism, lipid metabolism, folding, sorting, and degradation. The gut microbiota of the MCT-induced group was mainly involved in cellular processes and signaling, xenobiotics biodegradation and metabolism, cellular community prokaryotes, enzyme families, carbohydrate metabolism, metabolism of other amino acids, lipid metabolism, and glycan biosynthesis and metabolism. These results indicate that the changes in gut microbiota were different among the three PH models (Appendix A).

### 3.5. Functional Indicators of the Hypoxia-Induced PH Rat Plasma Metabolome

To characterize the plasma metabolome of Hyp-induced PH rats, we undertook untargeted metabolomic profiling of plasma samples and identified 215 metabolites. Principal component analysis (PCA) revealed a significant difference between the Nor and Hyp groups (Figure 6A). The heatmap showed that 41 metabolites were significantly different between the Nor and Hyp groups (Figure 6B and Appendix A). Compared with the Nor group, 35 were higher and 6 were lower in the Hyp group. Pathway enrichment analysis identified the following pathways as enriched in the Hyp group: ABC transporters; Ala and Asp metabolism; cyanoamino acid metabolism; Cys metabolism; glucosinolate biosynthesis; Gly, Ser, and Thr metabolism; His metabolism; pantothenate and CoA biosynthesis; protein digestion and absorption; valine, leucine, and Ile metabolism; valine, leucine and isoleucine biosynthesis, and beta-alanine metabolism (Appendix A). To identify the biomarkers of PH in plasma, we compared the plasma metabolites between the Nor group and Hyp group, and found that most high-impact metabolites were linked to PH: plasma levels of metabolites with consistently high contributions included the amino acids His, Ala, and Ser, and metabolites related to aliphatic polyamine, spermidine, and the ketone body 2-hydroxybutyric acid; also implicated were ADMA and cystathionine (Figure 6C).

### 3.6. Correlation Analysis of Microbial Diversity with Plasma Metabolites in Hypoxia-Induced PH Rats

As a bridge between the microbiome and host, metabolites of the gut microbiota can impact host physiological status both within the gut and after entering the bloodstream. Pathways for the synthesis of several amino acids were enriched (Appendix A), and elevated arginine coincided with the increased abundance of the arginine biosynthesis bacteria *Blautia* and *Bifidobacterium* in Hyp-induced PH rats (Figure 6B,D). The trimethylamine N-oxide- and trp-producing bacterium *Streptococcus* was also increased in Hyp-induced PH rats (Figure 6E).

The correlation analysis between the 75 bacteria identified above and the 53 metabolites revealed significant associations, many of which involved species enriched in Hyp-induced rats (Figure 7). Lactic acid was strongly positively correlated with *Lactococcus*, suggesting that the correlation analysis between metabolites and the gut microbiota is valid. Hypoxia-enriched His was positively correlated with *Bifidobacterium*, *Staphylococcus*, *Anaerovorax*, *Streptococcus*, Veillonella, and *unidentified Clostridiales*. Val was positively correlated with *Bifidobacterium*, *Anaerovorax*, and *Streptococcus*. Ser was positively correlated with *Staphylococcus*, *Anaerovorax*, *Streptococcus*, *Veillonella*, and *unidentified Clostridiales*. Arg was positively correlated with *Staphylococcus* and *Butyrivibrio*. Ile was positively correlated with *Anaerovorax* and *Streptococcus*. Gly was positively correlated with *Bifidobacterium*, *Staphylococcus*, *Romboutsia*, *Streptococcus*, *Veillonella*, *Butyrivibrio*, *Corynebacterium*, *Candidatus Arthromitus*, *Erysipelatoclostridium*, and *unidentified Clostridiales*. Nor rats enriched in 2-hydroxybutyric acid were strongly negatively correlated with *Veillonella* and *unidentified Christensenellaceae* (Figure 7).

## 4. Discussion

The gut microbiota consists of multiple microorganisms, including bacteria, fungi, parasites, and viruses. The host provides nutrients for these microorganisms, and the gut microbiota interacts with the host to maintain homeostasis. Gut microbiota dysbiosis is associated with various disease processes [30]. An imbalance in the gut microbiota has been observed in IBD, IPAH [13], CTEPH [31], and COPD patients [14]. In MCT- and HySu-induced PH rats, the gut microbiota was also disrupted [16,26,32]. Although all of the studies documented alterations in the gut microbiota, current data regarding the gut microbiota were collected at a fixed point in time and do not reflect the process of change. These studies are limited to uncovering the functions of gut microbiota in the development of PH. In this study, we collected the stool of rats under Hyp-, HySu-, and MCT-induction every 2 weeks and used 16S rRNA-seq to track the dynamic change in gut microbiota. The α-diversity and β-diversity of the gut microbiota changed during the development of rat PH models, suggesting that the gut microbiota is dynamic.

Compared with the Nor group, the α-diversity of the gut microbiota in HySu-induced rats was decreased at week 2, and this result is consistent with the change in gut microbiota in CTEPH [31] and IPAH patients [13]. These results further indicated that the HySu-induced rat PH model is similar to type 1 PAH. It is not clear why the α-diversity of the gut microbiota did not differ between HySu-induced rats and Nor group rats at week 4. Unexpectedly, the variation in α-diversity of the gut microbiota in Hyp-induced rats was similar to that in MCT-induced PH rats. MCT-induced PH is similar to type 1 PAH, and Hyp-induced PH is similar to type 3 PH. This is the first time that constancy in α-diversity of gut microbiota was found in both models.

The PCA showed that the samples from the Nor-, Hyp-induced, HySu-induced, and MCT-treated groups were clustered together within groups and groups were separated from each other, with each group containing a specific gut microbiota. These findings indicate that our system was reliable.

Some enriched bacteria can serve as biomarkers of disease; for example, *Erysipelotrichia* was positively correlated with tumor necrosis factor alpha (TNF) in patients who were infected with HIV [33,34]. *Alphaproteobacteria* can trigger autoimmune disease [35]. *Bifidobacteriaceae*, *Eubacteriaceae*, *Lactobacillaceae*, *Micrococcaceae*, *Streptococcaceae*, and *Veillonellaceae* were enriched in COPD [14]. Our previous study identified harmful *Erysipelotrichaceae*, *Alphaproteobacteria*, *Christensenella timonensis*, *Coriobacteriales*, *Rhodospirillales*, and *Eggerthellaceae* as biomarkers of Hyp-indu**c**ed PH mice [36]. In this system, *bacilli*, *Lactobacillales*, *Lactobacillaceae*, *Fusicatenibacter*, and *Staphylococcus sciuri* were predominant in the HySu-induced group. *Clostridiales*, Lachnospiraceae, *Acetatifactor*, Rikenellaceae, *Alistipes*, and *Adiercreutzia* were predominant in the MCT-induced group at week 2. *Turicibacter*, *Erysipelotrichaceae*, *Erysipelotrichia*, and *Erysipelotrichales* were identified as biomarkers in the MCT-induced group. *Unidentified Lachnospiraceae* were identified as biomarkers in the HySu-induced group. *Coryhebacterum urealyticum*, *Bacteroidetes*, *Bacteroidles*, *Bacteroidia*, and *Muribaculaceae* were identified as biomarkers in the Hyp-induced group (Figure 5E,F). These biomarkers may help to predict disease predisposition, activity, severity, and responsiveness to therapy. These results further suggest that the gut microbiota is dynamically changing at the different stages of PH.

An earlier study demonstrated that COPD patients have increased intestinal permeability [17,18]. The gut microbiota can influence immunity at distant sites. For example, *Lachnospiraceae* and *Ruminococcaceae* are short-chain fatty acid (SCFA)-producing bacteria [37]. SCFAs promote IL-22 production by CD4+ T cells and ILCs through G-protein receptor 41 (GPR41) and inhibition of histone deacetylase (HDAC) [38]. The SCFA butyrate can promote B10 cell generation by activating the p38 MAPK pathway [39]. The p38 MAPK pathway is activated in pulmonary hypertension [40,41,42]. We found an increased abundance of *Ruminococcaceae* in the HySu-induced group at week 4. *Lachnospiraceae* were increased in the MCT-induced and Hyp-induced groups at week 2, and both were stable from week 1 to week 4 in the Nor group. *Clostridioides* can lead to inflammation by producing two toxins [43]. Inflammation has been proven to induce PH. Our data indicated that *Clostridioides* was increased in the MCT-induced and Hyp-induced groups at week 4. These results indicate that the gut microbiota–SCFA–inflammation axis may promote the development of PH.

It is clear that the metabolites of gut microbiota can arrive at distant sites and induce disease, but how the metabolites of gut microbiota influence the lung is not well understood in PH. In this study, we screened PH-enriched plasma metabolites in Hyp-induced PH rats and identified 35 PH-enriched metabolites that originated from the gut microbiota.

Elevated arginine, proline, and ornithine are recognized as hallmarks of PAH pathogenesis [44]. Our study found that arginine, proline, and ornithine were elevated in the plasma of Hyp-induced PH rats. The abundances of the arginine biosynthesis bacteria *Blautia* and *Bifidobacterium* were increased in Hyp-induced PH rats. *Streptococcus* is a trimethylamine *N*-oxide biosynthetic bacteria. Both *Streptococcus* and trimethylamine *N*-oxide are increased in Hyp-induced PH rats. Trimethylamine *N*-oxide was reported to accelerate atherosclerosis. These results provide evidence that the gut-lung axis is mediated by metabolites. In addition, we used a correlation analysis to explore some new associations between the gut microbiota and metabolites (Figure 7). How these metabolites interact with the gut microbiota remains to be further studied.

## 5. Conclusions

In summary, our data indicate that the gut microbiota was disordered and dynamic changes occurred in Hyp-, HySu-, and MCT-induced PH rats. The change in α-diversity in Hyp-induced PH rats was similar to that in MCT-induced PH rats and different from that in HySu-induced PH rats. Six plasma biomarkers were identified in Hyp-induced PH rats. Finally, the correlation between the gut microbiota and metabolites were identified.

## Figures and Tables

**Figure 1 microorganisms-11-00472-f001:**
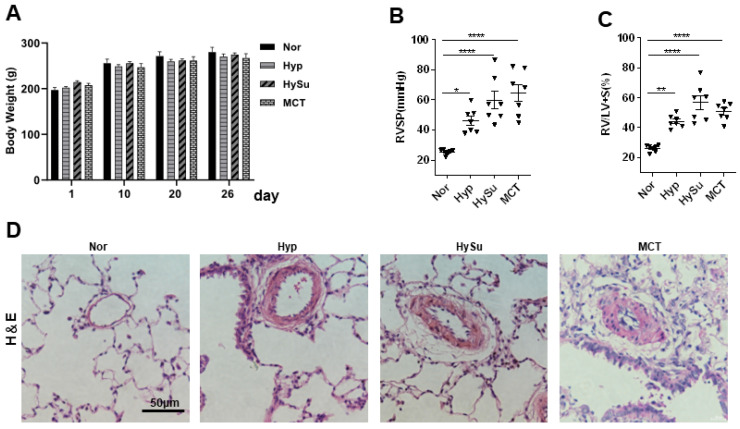
Hyp-, HySu-, and MCT-induced PH in rats. (**A**) The body weight of rats was measured at the indicated time points (*n* = 5–7). (**B**,**C**) RVSP and RV/LV+S ratio, respectively, in three PH rat models. Data are presented as the mean ± SEM. For multiple comparisons, one-way ANOVA (analysis of variance) for normally distributed samples followed by the Tukey HDS (honestly significant difference) method. (* *p* < 0.05, ** *p* < 0.01, **** *p* < 0.0001, *n* = 7). (**D**) Representative hematoxylin and eosin-stained lung sections in three PH rat models. Scale bars: 50 μm. Nor: normoxia group; Hyp: hypoxia-induced group; HySu: hypoxia/Sugen5416-induced group; MCT: MCT-induced group. 0, 2, and 4 represent the week of treatment; Filled triangle represent the rat individual in (**B**).

**Figure 2 microorganisms-11-00472-f002:**
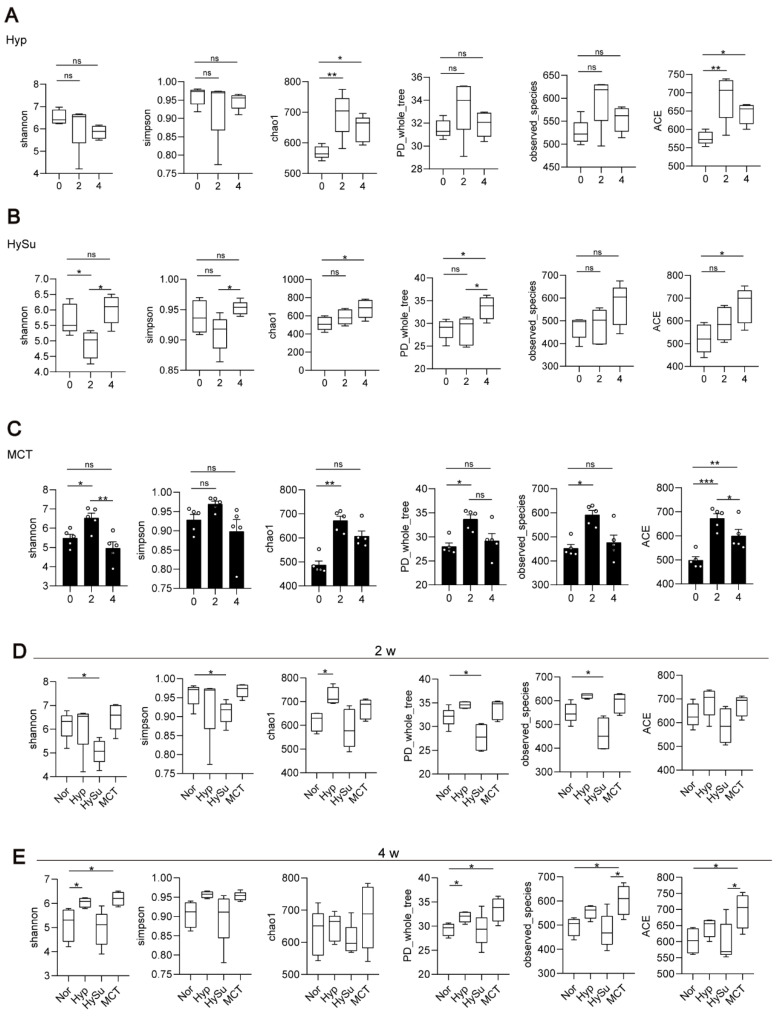
The dynamic α-diversity values and richness indices at week 2 and week 4 in Hyp-, HySu-, and MCT-induced PH rat models. Feces were collected from rats at weeks 0, 2, and 4. (**A**–**E**) The α-diversity and β-diversity of the gut microbiota in Hyp-, HySu-, and MCT-induced PH rats were measured by the Shannon, Simpson, Chao, PD whole tree, and observed species indices. Chao1 estimates the number of species, whereas Shannon estimates the effective number of species. Data are presented as the mean ± SEM of 5 animals, * *p* < 0.05, ** *p* < 0.01, *** *p* < 0.001, ns: no significance, and one-way ANOVA followed by Student–Newman–Keuls multiple comparison test for parametric distributions. The Kruskal–Wallis test followed by the Dunnett post hoc test were used for non-parametric distributions. Nor: normoxia group; Hyp: hypoxia-induced group; HySu: hypoxia/Sugen 5416-induced group; MCT: MCT-induced group. 0, 2, and 4 represent the week of treatment.

**Figure 3 microorganisms-11-00472-f003:**
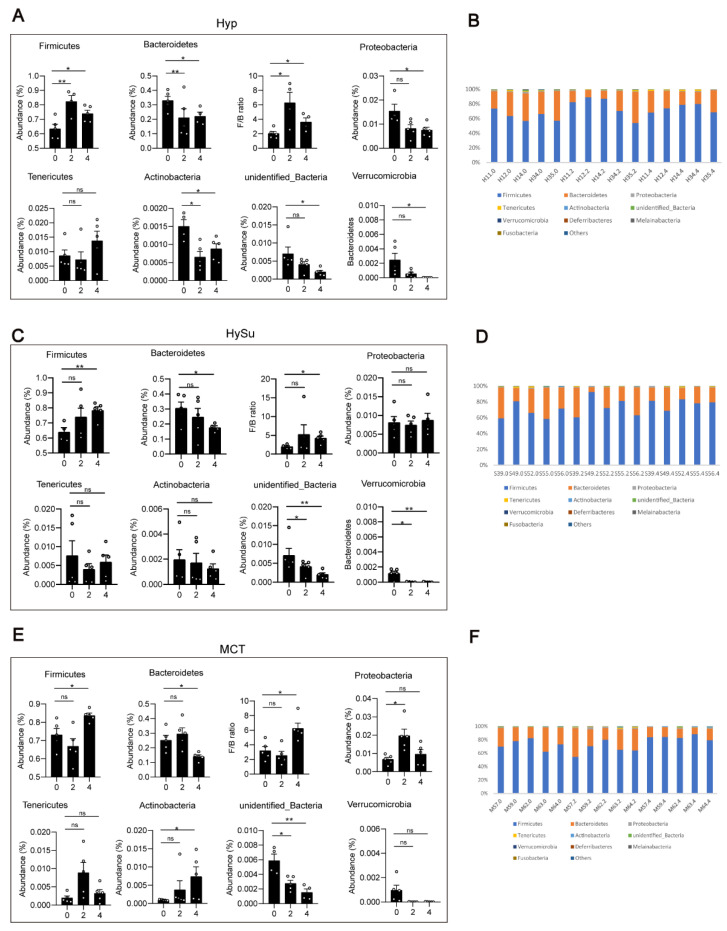
Fecal microbial composition at the phylum level. (**A**–**F**) The changes in *Firmicutes*, *Bacteroidetes*, *Proteobacteria*, *unidentified bacteria*, *Melainabateria*, and the ratio of *Firmicutes/Bacteroidetes* in Hyp- (**A**,**B**), HySu- (**C**,**D**), and MCT- (**E**,**F**) induced PH rats. Data are presented as the mean ± SEM, *n* = 4–5 per group; * *p* < 0.05, ** *p* < 0.01, ns: no significance, and one-way ANOVA followed by Student–Newman–Keuls multiple comparison test for parametric distributions. The Kruskal–Wallis test followed by the Dunnett post hoc test were used for non-parametric distributions. H11, H12, H14, H34, and H35 represent the codes of rats in the Hyp-induced group (Hyp). S39, S49, S52, S55, and S56 represent the codes of rats in the HySu-induced group (HySu). M57, M59, M62, M63, and M64 represent the codes of rats in the MCT-induced group (MCT). 0, 2, and 4 represent the week of treatment.

**Figure 4 microorganisms-11-00472-f004:**
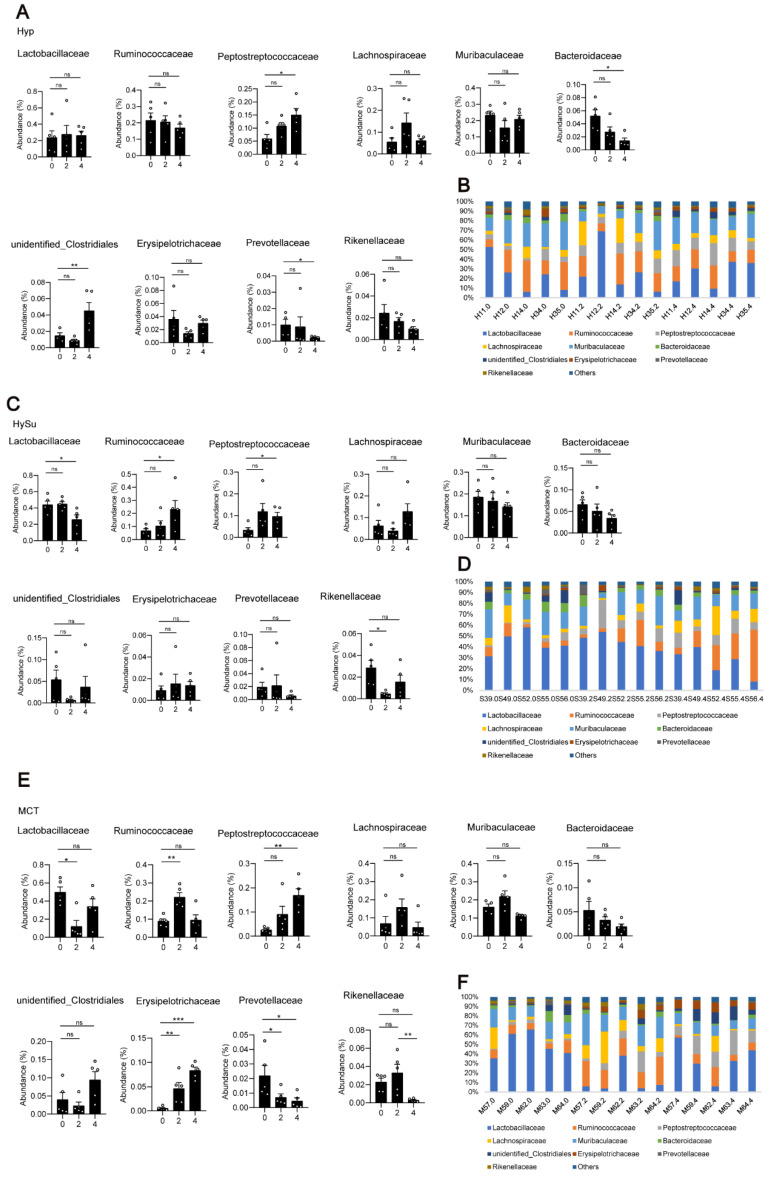
Fecal microbial composition at the family level. The most abundant taxa change at the family level in Hyp- (**A**,**B**), HySu- (**C**,**D**), and MCT- (**E**,**F**) induced PH rats. Data are presented as the mean ± SEM, *n* = 4–5 per group; * *p* < 0.05, ** *p* < 0.01, *** *p* < 0.001, ns: no significance, and one-way ANOVA followed by Student–Newman–Keuls multiple comparison test for parametric distributions. The Kruskal–Wallis test followed by the Dunnett post hoc test were used for non-parametric distributions. H11, H12, H14, H34, and H35 represent the codes of rats in the Hyp-induced group (Hyp). S39, S49, S52, S55, and S56 represent the codes of rats in the HySu-induced group (HySu). M57, M59, M62, M63, and M64 represent the codes of rats in the MCT-induced group (MCT). 0, 2, and 4 represent the week of treatment.

**Figure 5 microorganisms-11-00472-f005:**
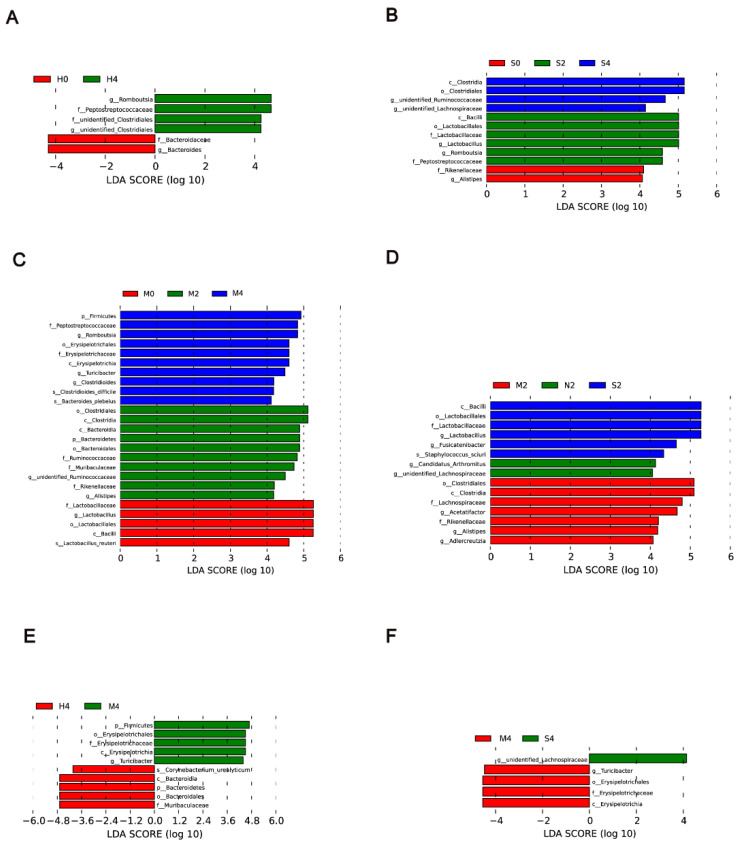
LDA of the microbial community variation. (**A**–**C**) A linear discriminant effect size (LEfSe) analysis was performed (alpha value ≥ 0.05, logarithmic LDA score threshold ≥ 2) at weeks 0, 2, and 4 in the Hyp group (**A**), HySu group (**B**), and MCT group (**C**). (**D**) LEfSe analysis was performed (alpha value ≥ 0.05, logarithmic LDA score threshold ≥ 2) on Nor, Hyp, HySu, and MCT groups at week 2. (**E**) LEfSe analysis was performed (alpha value ≥ 0.05, logarithmic LDA score threshold ≥ 2) on Nor, Hyp, and MCT at week 4. (**F**) LEfSe analysis was performed (alpha value ≥ 0.05, logarithmic LDA score threshold ≥ 2) on Nor, Hyp, and HySu at week 4. Nor (N): normoxia group; Hyp (H): hypoxia-induced group; HySu (S): hypoxia/Sugen 5416-induced group; MCT (M): MCT-induced group. 0, 2, and 4 represent the week of treatment.

**Figure 6 microorganisms-11-00472-f006:**
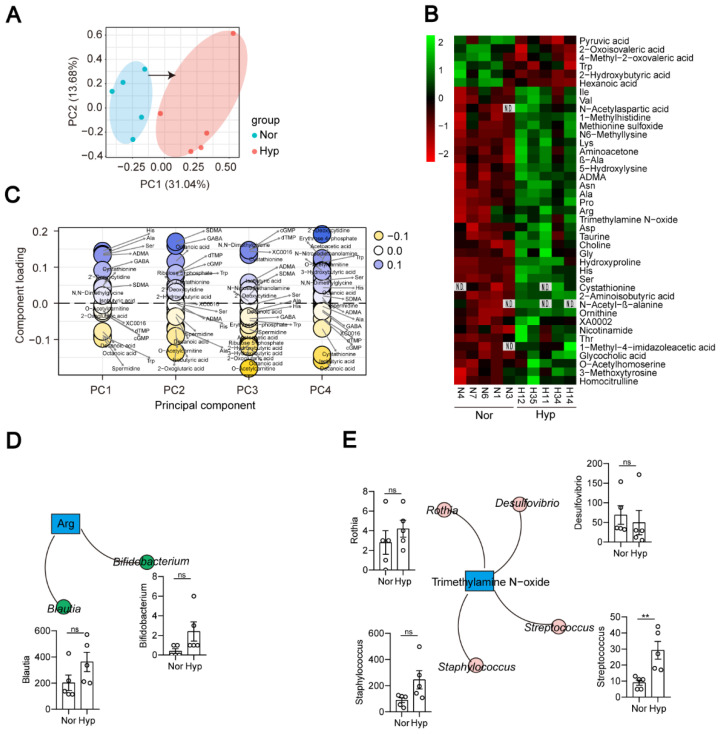
Metabolites of Hyp-induced PH rats (*n* = 5) are distinguished from those of normoxia rats (*n* = 5). (**A**) PCA scatter plot of the metabolite profile. (**B**) Heatmap analysis of differential metabolites between the Nor and Hyp groups. N1, N3, N4, N6, and N7 represent the codes of rats in the normoxia group (Nor). H11, H12, H14, H34, and H35 represent the codes of rats in the Hyp-induced group (Hyp). (**C**) Identification of metabolite biomarkers. (**D**,**E**) Correlations between bacterial species and metabolites. Data are presented as the mean ± SEM. Statistical significance was evaluated by Student’s *t* test (** *p* < 0.01, ns: no significance, *n* = 5).

**Figure 7 microorganisms-11-00472-f007:**
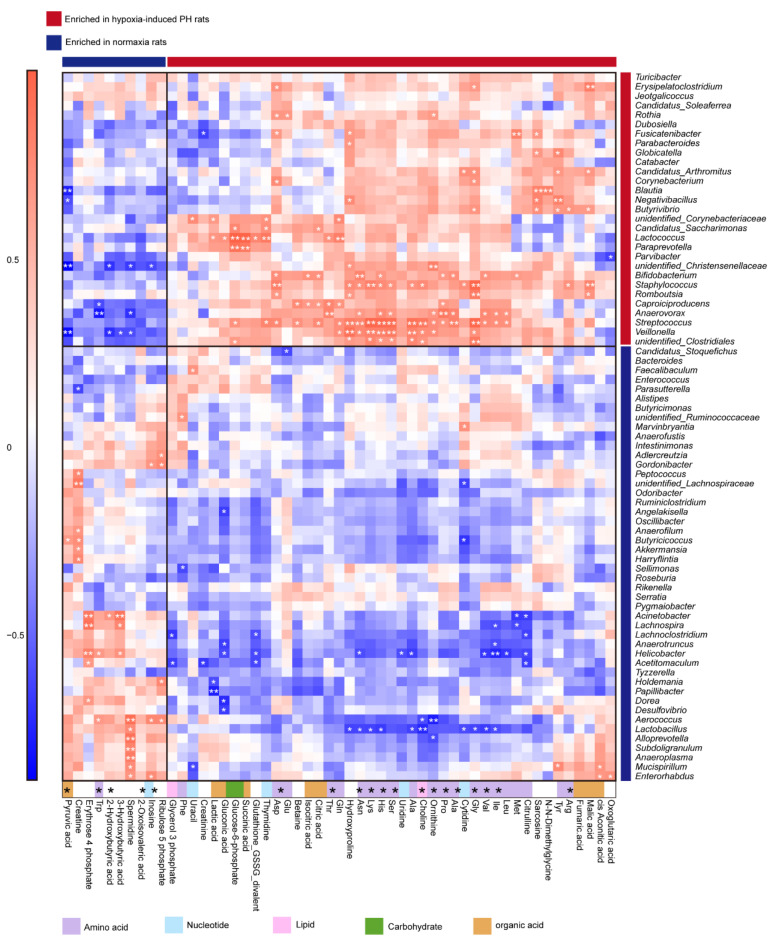
The correlation between gut microbiota and metabolites in normoxia- and hypoxia-induced rats. Heatmap of Spearman correlations between the bacteria whose abundances significantly changed in the Hyp (*n* = 5) vs. Nor (*n* = 5) groups and the metabolites with important functions and significant differences. Enrichment in either group indicated by colored bars to the right and top of the plot, blue bands represent enrichment in Nor group, red bands represent enrichment in Hyp group. Super pathway of metabolites is indicated by colored bars. Significant correlations denoted by white stars (* *p* < 0.05; ** *p* < 0.01, Student’s *t* test (two-sided), Benjamini–Hochberg adjustment for multiple comparisons. Exact *p* values are provided in Appendix A). Nor: normoxia group; Hyp (H): hypoxia-induced group.

## Data Availability

Not applicable.

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
