# Peer review of "Gut Microbiota and Metabolome Changes in Three Pulmonary Hypertension Rat Models"

_microorganisms, 2023, doi:10.3390/microorganisms11020472_

Round 1

Reviewer 1 Report

Manuscript microorganisms-2184750 deals with the association of different gut bacteria with specific metabolites in three different pulmonary rat models under hypertension conditions. Given that the effort was mainly to associate the gut bacteria with metabolites, the manuscript falls within the general concept of Microorganisms journal.

The design of the study is very good and the data support the hypothesis driven in the study. It is of great importance to associate metabolites with potential disorders even if rats are used; There might be a good evidence also  for humans. Metabolomics have greatly contributed to these demands. The authors used capillary electrophoresis time-of-flight mass spectrometry (CE-TOFMS). They must briefly describe in the relevant section the strengths and limitations of this technique.

In addition, they must try to avoid the conscutive use of first person (we) throughout the text and rephrase the relevant sections.

I have indicated within the attached pdf, the main problems of this work. 

Finally, I believe that this study would benefit the readers and research community. Therefore, I suggest its publication after a minor and thorough revision regardind the data interpretation without the consecutive use of first person.

Reviewer 2 Report

This is a well written manuscript. Below is my feedback.

Introduction

The authors do a very good job with the introduction however, instead of providing the readers with the results and the analysis at the end the authors should provide a hypothesis while linking it back to their literature review.

Methodology

I have several questions regarding the methodology

1. I am assuming that you had a cutoff for what PC was considered significant. 

2. Why did you use a Student's T-test? With N=5 per group it's difficult to justify a parametric test. Further because of the fact that you were not able to complete a mixed ANOVA did you correct for mulitple tests?

3. It's only when I get to your results that I am now realizing that you're using the PCAs to determine if the microbiota grouped different between groups. Please identify that in your methodology.

4. Looking at the distribution of your data in Figure 2, I think you should use non-parametric analyses because your sample size is too small and your data distribution is too wide.

5. You should also write in your methodology that you performed clustering analyses.

Results

1. Figure 7- which one was the norm vs hypoxia

2. What was the cutoff point for the PCs? 

3. With this many analyses there is a high risk for Type I errors

With the number of potential statistical errors, I cannot properly evaluate the discussion section.

Round 2

Reviewer 2 Report

Thank you for taking the time to respond to my questions.